# AnyV2V: A Tuning-Free Framework For Any Video-to-Video Editing Tasks

♠†**Max Ku**\*, ♠†**Cong Wei**\*, ♠†**Weiming Ren**\*, ♡**Harry Yang**, ♠†**Wenhu Chen**
♠**University of Waterloo**, †**Vector Institute**, ♡**Harmony.AI**
`{m3ku, c58wei, w2ren, wenhuchen}@uwaterloo.ca`

Reviewed on OpenReview: `https://openreview.net/forum?id=RFrJCkw2oa`

## Abstract

In the dynamic field of digital content creation using generative models, state-of-the-art video editing models still do not offer the level of quality and control that users desire. Previous works on video editing either extended from image-based generative models in a zero-shot manner or necessitated extensive fine-tuning, which can hinder the production of fluid video edits. Furthermore, these methods frequently rely on textual input as the editing guidance, leading to ambiguities and limiting the types of edits they can perform. Recognizing these challenges, we introduce AnyV2V, a novel tuning-free paradigm designed to simplify video editing into two primary steps: (1) employing an off-the-shelf image editing model to modify the first frame, (2) utilizing an existing image-to-video generation model to generate the edited video through temporal feature injection. AnyV2V can leverage any existing image editing tools to support an extensive array of video editing tasks, including prompt-based editing, reference-based style transfer, subject-driven editing, and identity manipulation, which were unattainable by previous methods. AnyV2V can also support any video length. Our evaluation shows that AnyV2V achieved CLIP-scores comparable to other baseline methods. Furthermore, AnyV2V significantly outperformed these baselines in human evaluations, demonstrating notable improvements in visual consistency with the source video while producing high-quality edits across all editing tasks. The code is available at https://github.com/TIGER-AI-Lab/AnyV2V.

## 1 Introduction

The development of deep generative models (Ho et al., 2020) has led to significant advancements in content creation and manipulation, especially in digital images (Rombach et al., 2022; Nichol et al., 2022; Brooks et al., 2023; Ku et al., 2024; Chen et al., 2023c; Li et al., 2023). However, video generation and editing have not reached the same level of advancement as images (Wang et al., 2023; Chen et al., 2023a; 2024; Ho et al., 2022). In the context of video editing, training a large-scale video editing model presents considerable challenges due to the scarcity of paired data and the substantial computational resources required.

To overcome these challenges, researchers proposed various approaches (Geyer et al., 2023; Cong et al., 2023; Wu et al., 2023a;b; Liu et al., 2023a; Liang et al., 2023; Gu et al., 2023b; Cong et al., 2023; Zhang et al., 2023d; Qi et al., 2023; Ceylan et al., 2023; Yang et al., 2023; Jeong & Ye, 2023; Guo et al., 2023), which can be categorized into two types: (1) zero-shot adaptation from pre-trained text-to-image (T2I) models or (2) fine-tuned motion module from a pre-trained T2I or text-to-video (T2V) models. The zero-shot methods (Geyer et al., 2023; Cong et al., 2023; Jeong & Ye, 2023; Zhang et al., 2023d) usually suffer from flickering issues due to a lack of temporal understanding. On the other hand, fine-tuning methods (Wu et al., 2023a; Chen et al., 2023b; Wu et al., 2023b; Gu et al., 2023b; Guo et al., 2023) require more time and computational overhead to edit videos. Moreover, all the methods can only adhere to certain types of edits. For example, a user might want to perform edits on visuals that are out-of-distribution from the learned text encoder (e.g. change a person to a character from their artwork). Thus, a highly customizable

solution is more desired in video editing applications. It would be ideal if there were methods that allowed for a seamless combination of human artistic input and the assistance provided by AI, allowing for a synergy between human effort and AI, maintaining creators' creativity while producing high-quality outputs.

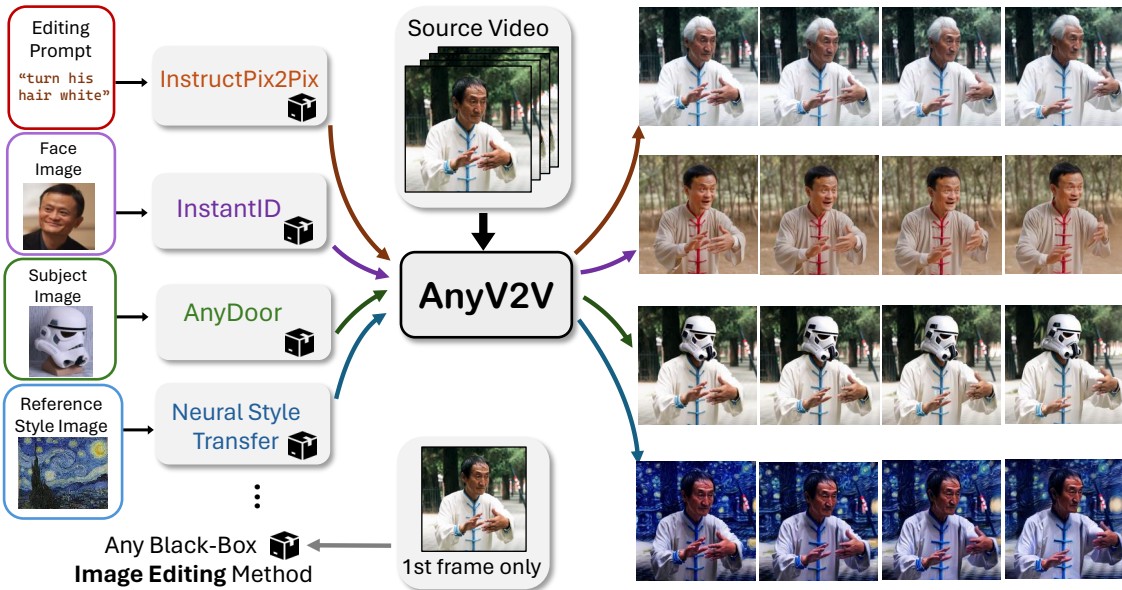

Figure 1: AnyV2V is an evolving framework to handle all types of video-to-video editing tasks without any parameter tuning. AnyV2V disentangles video editing into two simpler problems: (1) Single image editing and (2) Image-to-video generation with video reference.

With the above motivations, we aim to develop a video editing framework that requires no fine-tuning and caters to user demand. In this work, we present AnyV2V, designed to enhance the controllability of zero-shot video editing by decomposing the task into two pivotal stages:

1. Apply image edit on the first frame with any off-the-shelf image editing model or human effort.

2. Leverage the innate knowledge of the image-to-video (I2V) model to generate the edited video with the edited first frame, source video latent, and the intermediate temporal features.

Our objective is to propagate the edited first frame across the entire video while ensuring alignment with the source video. To achieve this, we employ I2V models (Zhang et al., 2023c; Chen et al., 2023d; Ren et al., 2024) for DDIM inversion to enable first-frame conditioning. With the inverted latents as initial noise and the modified first frame as the conditional signal, the I2V model can generate videos that are not only faithful to the edited first frame but also follow the appearance and motion of the source video. To further enforce the consistency of the appearance and motion with the source video, we perform feature injection in the I2V model. The two-stage editing process effectively offloads the editing operation to existing image editing tools. This design (detail in Section 4) helps AnyV2V excel in:

- **Compatibility**: It provides a highly customized interface for a user to perform video edits on any modality. It can seamlessly integrate any image editing methods to perform diverse editing.

- **Simplicity**: It does not require any fine-tuning nor additional video features like previous works (Gu et al., 2023b; Wu et al., 2023b; Ouyang et al., 2023) to achieve high appearance and temporal consistency for video editing tasks.

From our findings, without any fine-tuning, AnyV2V can perform video editing tasks beyond the scope of current publicly available methods, such as reference-based style transfer, subject-driven editing, and identity manipulation. AnyV2V also can perform prompt-based editing and achieve superior results on common video editing evaluation metrics compared to the baseline models (Geyer et al., 2023; Cong et al., 2023; Wu et al., 2023b). We show both quantitatively and qualitatively that our method outperforms existing SOTA

baselines in Section 5.3 and Appendix B. AnyV2V is favoured in 69.7% of samples for prompt alignment and 46.2% overall preference in human evaluation, while the best baseline only achieves 31.7% and 20.7%, respectively. Our method also reaches the highest CLIP-Text score of 0.2932 in text alignment and a competitive CLIP-Image score of 0.9652 in temporal consistency.

All these achievements are thanks to the AnyV2V's design to harness the power of off-the-shelf image editing models from advanced image editing research. Through a comprehensive study and evaluation of the effectiveness of our design, our key observation is that the inverted noise latent and feature injection serve as critical components to guide the video motion, and the I2V model itself has good capabilities in generating motions. We also found that by inverting long videos exceeding the I2V models' training frames, the inverted latents enable the I2V model to produce longer videos, making long video editing possible. To summarize, The main contributions of our work are three-fold:

- We proposed AnyV2V as a first fundamentally different solution for video editing, treating video editing as a simpler image editing problem.

- We showed that AnyV2V can support long video editing by inverting videos that extend beyond the training frame lengths of I2V models.

- Our extensive experiments showcased the superior performance of AnyV2V when compared to the existing SOTA methods, highlighting the potential of leveraging I2V models for video editing.

## 2  Related Works

Video generation has attracted considerable attention within the field (Chen et al., 2023a; 2024; OpenAI; Wang et al., 2023; Hong et al., 2022; Zhang et al., 2023a; Henschel et al., 2024; Wang et al., 2024c; Xing et al., 2023; Chen et al., 2023d; Bar-Tal et al., 2024; Ren et al., 2024; Zhang et al., 2023c). However, video manipulation also represents a significant and popular area of interest. Initial attempts, such as Tune-A-Video (Wu et al., 2023b) and VideoP2P (Liu et al., 2023b) involved fine-tuning a text-to-image model to achieve video editing by learning the continuous motion. The concurrent works at that time such as Pix2Video (Ceylan et al., 2023) and Fate-Zero (Qi et al., 2023) go for zero-shot approach, which leverages the inverted latent from a text-to-image model to retain both structural and motion information. The progressively propagates to other frames edits. Subsequent developments have enhanced the results but generally follow the two paradigms (Geyer et al., 2023; Wu et al., 2023a; Cong et al., 2023; Yang et al., 2023; Ceylan et al., 2023; Ouyang et al., 2023; Guo et al., 2023; Gu et al., 2023b; Esser et al., 2023; Chen et al., 2023b; Jeong & Ye, 2023; Zhang et al., 2023d; Cheng et al., 2023). Control-A-Video (Chen et al., 2023b) and ControlVideo (Zhang et al., 2023d) leveraged ControlNet (Zhang et al., 2023b) for extra spatial guidance. TokenFlow (Geyer et al., 2023) leveraged the nearest neighbor field and inverted latent to achieve temporally consistent edit. Fairy (Wu et al., 2023a) followed both paradigms which they fine-tuned a text-to-image model and also cached the attention maps to propagate the frame edits. VideoSwap (Gu et al., 2023b) requires additional parameter tuning and video feature extraction (e.g. tracking, point correspondence, etc) to ensure appearance and temporal consistency. CoDeF (Ouyang et al., 2023) allows the first image edit to propagate the other frames with one-shot tuning. UniEdit (Bai et al., 2024) leverages the inverted latent and feature maps injection to achieve a wide range of video editing with a pre-trained text-to-video model (Wang et al., 2023).

However, none of the methods can offer precise control to users, as the edits may not align with the user's exact intentions or desired level of detail, often due to the ambiguity of natural language and the constraints of the model's capabilities. For example, VideoP2P (Liu et al., 2023a) is restricted to only word-swapping prompts due to the reliance on cross-attention. There is a clear need for a more precise and comprehensive solution for video editing tasks. Our work AnyV2V is the first work to empower a diverse array of video editing tasks. We compare AnyV2V with the existing methods in Table 1. As can be seen, our method excels in its applicability and compatibility.

Table 1: Comparison with different video editing methods and the type of editing tasks.

| Method | Prompt-based Editing | Reference-based Style Transfer | Subject-Driven Editing | Identity Manipulation | Tuning-Free? | Backbone |
|---|---|---|---|---|---|---|
| Tune-A-Video (Wu et al., 2023b) | ✓ | ✗ | ✗ | ✗ | ✗ | Stable Diffusion |
| Pix2Video (Ceylan et al., 2023) | ✓ | ✗ | ✗ | ✗ | ✓ | SD-Depth |
| Gen-1 (Esser et al., 2023) | ✓ | ✗ | ✗ | ✗ | ✗ | Stable Diffusion |
| TokenFlow (Geyer et al., 2023) | ✓ | ✗ | ✗ | ✗ | ✓ | Stable Diffusion |
| FLATTEN (Cong et al., 2023) | ✓ | ✗ | ✗ | ✗ | ✓ | Stable Diffusion |
| Fairy (Wu et al., 2023a) | ✓ | ✗ | ✗ | ✗ | ✓ | Stable Diffusion |
| ControlVideo (Zhang et al., 2023d) | ✓ | ✗ | ✗ | ✗ | ✗ | ControlNet |
| CoDeF (Ouyang et al., 2023) | ✓ | ✗ | ✗ | ✗ | ✗ | ControlNet |
| VideoSwap (Gu et al., 2023b) | ✗ | ✗ | ✓ | ✗ | ✗ | AnimateDiff |
| UniEdit (Bai et al., 2024) | ✓ | ✗ | ✗ | ✗ | ✓ | Any T2V Models |
| **AnyV2V (Ours)** | ✓ | ✓ | ✓ | ✓ | ✓ | Any I2V Models |

# 3 Preliminary

## 3.1 Image-to-Video (I2V) Generation Models

In this work, we focus on leveraging latent diffusion-based (Rombach et al., 2022) I2V generation models for video editing. Given an input first frame $I_1$, a text prompt $\mathbf{s}$ and a noisy video latent $\mathbf{z}_t$ at time step $t$, I2V generation models recover a less noisy latent $\mathbf{z}_{t-1}$ using a denoising model $\epsilon_\theta(\mathbf{z}_t, I_1, \mathbf{s}, t)$ conditioned on both $I_1$ and $\mathbf{s}$. The denoising model $\epsilon_\theta$ contains a set of spatial and temporal self-attention layers, where the self-attention operation can be formulated as:

$$Q = W^Q z, K = W^K z, V = W^V z, \tag{1}$$

$$\text{Attention}(Q, K, V) = \text{Softmax}(\frac{QK^\top}{\sqrt{d}})V, \tag{2}$$

where $z$ is the input hidden state to the self-attention layer and $W^Q$, $W^K$ and $W^V$ are learnable projection matrices that map $z$ onto query, key and value vectors, respectively. For spatial self-attention, $z$ represents a sequence of spatial tokens from each frame. For temporal self-attention, $z$ is composed of tokens located at the same spatial position across all frames.

## 3.2 DDIM Inversion

The denoising process for I2V generation models from $\mathbf{z}_t$ to $\mathbf{z}_{t-1}$ can be achieved using the DDIM (Song et al., 2020) sampling algorithm. The reverse process of DDIM sampling, known as DDIM inversion (Mokady et al., 2023; Dhariwal & Nichol, 2021), allows obtaining $\mathbf{z}_{t+1}$ from $\mathbf{z}_t$ such that $\mathbf{z}_{t+1} = \sqrt{\frac{\alpha_{t+1}}{\alpha_t}}\mathbf{z}_t + (\sqrt{\frac{1}{\alpha_{t+1}} - 1} - \sqrt{\frac{1}{\alpha_t} - 1}) \cdot \epsilon_\theta(\mathbf{z}_t, x_0, \mathbf{s}, t)$, where $\alpha_t$ is derived from the variance schedule of the diffusion process.

## 3.3 Plug-and-Play (PnP) Diffusion Features

Tumanyan et al. (2023a) proposed PnP diffusion features for image editing, based on the observation that intermediate convolution features $f$ and self-attention scores $A = \text{Softmax}(\frac{QK^\top}{\sqrt{d}})$ in a text-to-image (T2I) denoising U-Net capture the semantic regions (e.g. legs or torso of a human body) during the image generation process.

Given an input source image $I^S$ and a target prompt $P$, PnP first performs DDIM inversion to obtain the image's corresponding noise $\{\mathbf{z}_t^S\}_{t=1}^T$ at each time step $t$. It then collects the convolution features $\{f_t^l\}$ and attention scores $\{A_t^l\}$ from some predefined layers $l$ at each time step $t$ of the backward diffusion process $\mathbf{z}_{t-1}^S = \epsilon_\theta(\mathbf{z}_t^S, \varnothing, t)$, where $\varnothing$ denotes the null text prompt during denoising.

To generate the edited image $I^*$, PnP starts from the initial noise of the source image (i.e. $\mathbf{z}_T^* = \mathbf{z}_T^S$) and performs feature injection during denoising: $\mathbf{z}_{t-1}^* = \epsilon_\theta(\mathbf{z}_t^*, P, t, \{f_t^l, A_t^l\})$, where $\epsilon_\theta(\cdot, \cdot, \cdot, \{f_t^l, A_t^l\})$ represents

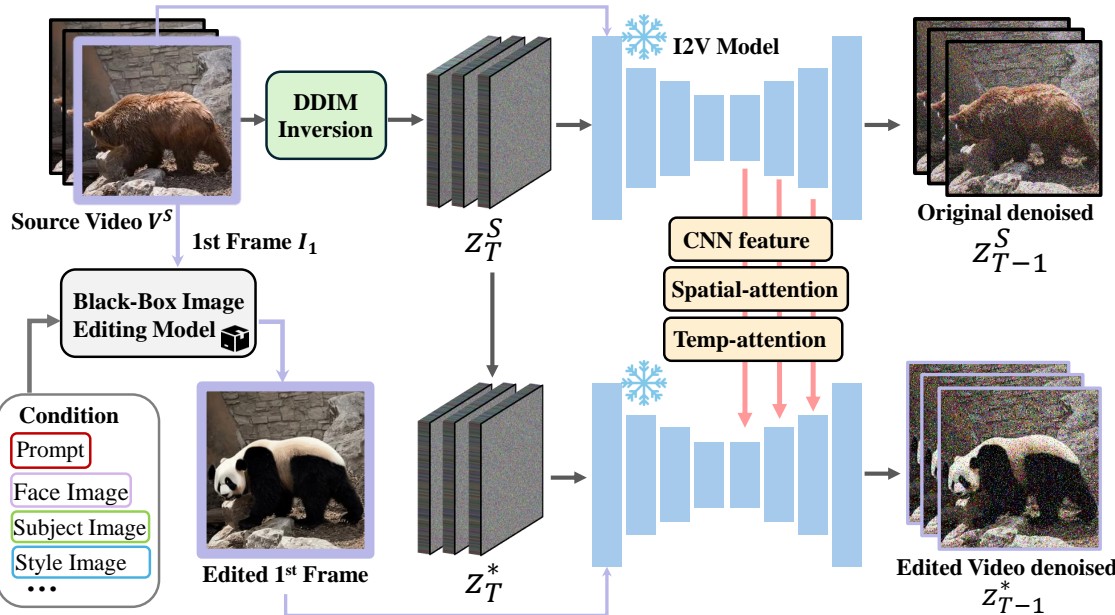

Figure 2: AnyV2V takes a source video $V^S$ as input. In the first stage, we apply a block-box image editing method on the first frame $I_1$ according to the editing task. In the second stage, the source video is inverted to initial noise $z_T^S$, which is then denoised using DDIM sampling. During the sampling process, we extract spatial convolution, spatial attention, and temporal attention features from the I2V models' decoder layers. To generate the edited video, we perform a DDIM sampling by fixing $z_T^*$ as $z_T^T$ and use the edited first frame as the conditional signal. During sampling, we inject the features and attention into corresponding layers of the model.

the operation of replacing the intermediate feature and attention scores $\{f_t^{l*}, A_t^{l*}\}$ with $\{f_t^l, A_t^l\}$. This feature injection mechanism ensures $I^*$ to preserve the layout and structure from $I^S$ while reflecting the description in $P$. To control the feature injection strength, PnP also employs two thresholds $\tau_f$ and $\tau_A$ such that the feature and attention scores are only injected in the first $\tau_f$ and $\tau_A$ denoising steps. Our method extends this feature injection mechanism to I2V generation models, where we inject features in convolution, spatial, and temporal attention layers. We show the detailed design of AnyV2V in Section 4.

## 4 AnyV2V

Our method presents a two-stage approach to video editing. Given a source video $V^S = \{I_1, I_2, I_3, ..., I_n\}$, where $I_i$ is the frame at time $i$ and $n$ denotes the video length, we first extract the initial frame $I_1$ and pass it into an image editing model $\phi_{\text{img}}$ to obtain an edited first frame $I_1^* = \phi_{\text{img}}(I_1, C)$. $C$ is the auxiliary conditions for image editing models such as text prompt, mask, style, etc. In the second stage, we feed the edited first frame $I_1^*$ and a target prompt $\mathbf{s}^*$ into an I2V generation model $\epsilon_\theta$ and employ the inverted latent from the source video $V^S$ to guide the generation process such that the edited video $V^*$ follows the motion of the source video $V^S$, the semantic information represented in the edited first frame $I_1^*$, and the target prompt $\mathbf{s}^*$. An overall illustration of our video editing pipeline is shown in Figure 2. In this section, we explain each core component of our method.

### 4.1 Flexible First Frame Editing

In visual manipulation, controllability is a key element in performing precise editing. AnyV2V enables more controllable video editing by utilizing image editing models to modify the video's first frame. This strategic approach enables highly accurate modifications in the video and is compatible with a broad spectrum of image editing models, including other deep learning models that can perform image style transfer (Gatys et al., 2015; Ghiasi et al., 2017; Lötzsch et al., 2022; Wang et al., 2024a), mask-based image editing (Nichol et al.,

2022; Avrahami et al., 2022), image inpainting (Suvorov et al., 2021; Ku et al., 2022), identity-preserving image editing (Wang et al., 2024b), and subject-driven image editing (Chen et al., 2023c; Li et al., 2023; Gu et al., 2023a).

## 4.2 Structural Guidance using DDIM Inverison

To ensure the generated videos from the I2V generation model follow the general structure as presented in the source video, we employ DDIM inversion to obtain the latent noise of the source video at each time step $t$. Specifically, we perform the inversion *without* text prompt condition but *with* the first frame condition. Formally, given a source video $V^S = \{I_1, I_2, I_3, ..., I_n\}$, we obtain the inverted latent noise for time step $t$ as:

$$\mathbf{z}_t^S = \text{DDIM\_Inv}(\epsilon_\theta(\mathbf{z}_{t+1}, I_1, \varnothing, t)), \tag{3}$$

where $\text{DDIM\_Inv}(\cdot)$ denotes the DDIM inversion operation as described in Appendix 3. In ideal cases, the latent noise $\mathbf{z}_T^S$ at the final time step $T$ (initial noise of the source video) should be used as the initial noise for sampling the edited videos. In practice, we find that due to the limited capability of certain I2V models, the edited videos denoised from the last time step are sometimes distorted. Following Li et al. (2023), we observe that starting the sampling from a previous time step $T' < T$ can be used as a simple workaround to fix this issue.

## 4.3 Appearance Guidance via Spatial Feature Injection

Our empirical observation (Section 5.4) suggests that I2V generation models already have some editing capabilities by only using the edited first frame and DDIM inverted noise as the model input. However, we find that this simple approach is often unable to correctly preserve the background in the edited first frame and the motion in the source video, as the conditional signal from the source video encoded in the inverted noise is limited.

To enforce consistency with the source video, we perform feature injection in both convolution layers and spatial attention layers in the denoising U-Net. During the video sampling process, we simultaneously denoise the source video using the previously collected DDIM inverted latents $\mathbf{z}_t^S$ at each time step $t$ such that $\mathbf{z}_{t-1}^S = \epsilon_\theta(\mathbf{z}_t^S, I_1, \varnothing, t)$. We preserve two types of features during source video denoising: convolution features $f^{l_1}$ before skip connection from the $l_1^{\text{th}}$ residual block in the U-Net decoder, and the spatial self-attention scores $\{A_s^{l_2}\}$ from $l_2 = \{l_{low}, l_{low+1}, ..., l_{high}\}$ layers. We collect the queries $\{Q_s^{l_2}\}$ and keys $\{K_s^{l_2}\}$ instead of directly collecting $A_s^{l_2}$ as the attention score matrices are parameterized by the query and key vectors. We then replace the corresponding features during denoising the edited video in both the normal denoising branch and the negative prompt branch for classifier-free guidance (Ho & Salimans, 2022). We use two thresholds $\tau_{conv}$ and $\tau_{sa}$ to control the convolution and spatial attention injection to only happen in the first $\tau_{conv}$ and $\tau_{sa}$ steps during video sampling.

## 4.4 Motion Guidance through Temporal Feature Injection

The spatial feature injection mechanism described in Section 4.3 significantly enhances the background and overall structure consistency of the edited video. While it also helps maintain the source video motion to some degree, we observe that the edited videos will still have a high chance of containing incorrect motion compared to the source video. On the other hand, we notice that I2V generation models, or video diffusion models in general, are often initialized from pre-trained T2I models and continue to be trained on video data. During the training process, parameters in the spatial layers are often frozen or set to a lower learning rate such that the pre-trained weights from the T2I model are less affected, and the parameters in the temporal layers are more extensively updated during training. Therefore, it is likely that a large portion of the motion information is encoded in the temporal layers of the I2V generation models. Concurrent work (Bai et al., 2024) also observes that features in the temporal layers show similar characteristics with optical flow (Horn & Schunck, 1981), a pattern that is often used to describe the motion of the video.

To better reconstruct the source video motion in the edited video, we propose to also inject the temporal attention features in the video generation process. Similar to spatial attention injection, we collect the source

*Prompt-based Editing*

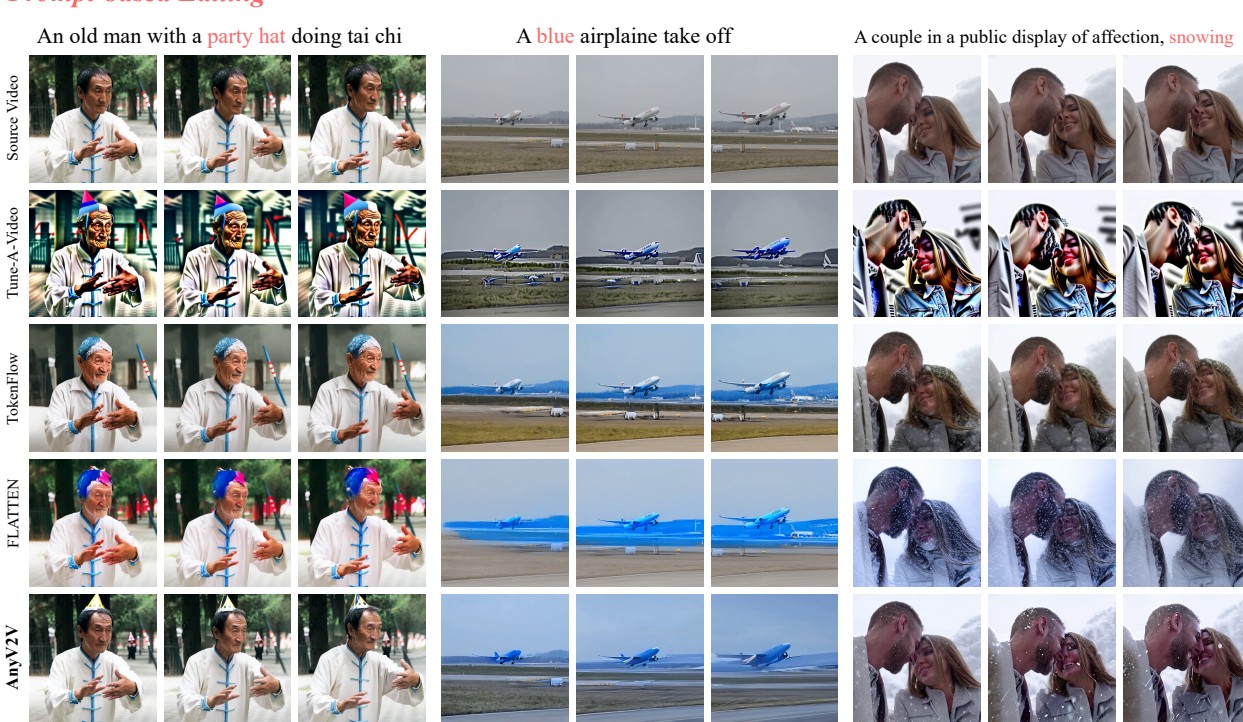

Figure 3: AnyV2V is robust in a wide range of prompt-based editing tasks while preserving the background. The results align the most with the text prompt and maintain high motion consistency.

video temporal self-attention queries $Q_t^{l_3}$ and keys $K_t^{l_3}$ from some U-Net decoder layers represented by $l_3$ and inject them into the edited video denoising branches. We also only apply temporal attention injection in the first $\tau_{ta}$ steps during sampling.

### 4.5 Putting it Together

Overall, combining the spatial and temporal feature injection mechanisms, we replace the editing branch features $\{f^{*l_1}, Q_s^{*l_2}, K_s^{*l_2}, Q_t^{*l_3}, K_t^{*l_3}\}$ with the features from the source denoising branch:

$$\mathbf{z}_{t-1}^* = \epsilon_\theta(\mathbf{z}_t^*, I^*, \mathbf{s}^*, t \; ; \{f^{l_1}, Q_s^{l_2}, K_s^{l_2}, Q_t^{l_3}, K_t^{l_3}\}), \tag{4}$$

where $\epsilon_\theta(\cdot \; ; \{f^{l_1}, Q_s^{l_2}, K_s^{l_2}, Q_t^{l_3}, K_t^{l_3}\})$ denotes the feature replacement operation across different layers $l_1, l_2, l_3$. Our proposed spatial and temporal feature injection scheme enables tuning-free adaptation of I2V generation models for video editing. Our experimental results demonstrate that each component in our design is crucial to the accurate editing of source videos. We showcase more qualitative results for the effectiveness of our model components in Section 5.

## 5 Experiments

### 5.1 Implementation Details

We employ AnyV2V on three off-the-shelf I2V generation models: I2VGen-XL[1] (Zhang et al., 2023c), ConsistI2V (Ren et al., 2024) and SEINE (Chen et al., 2023d). For all I2V models, we use $\tau_{conv} = 0.2T$, $\tau_{sa} = 0.2T$ and $\tau_{ta} = 0.5T$, where $T$ is the total number of sampling steps. We use the DDIM (Song et al.,

---

[1]We use the version provided in `https://huggingface.co/ali-vilab/i2vgen-xl`.

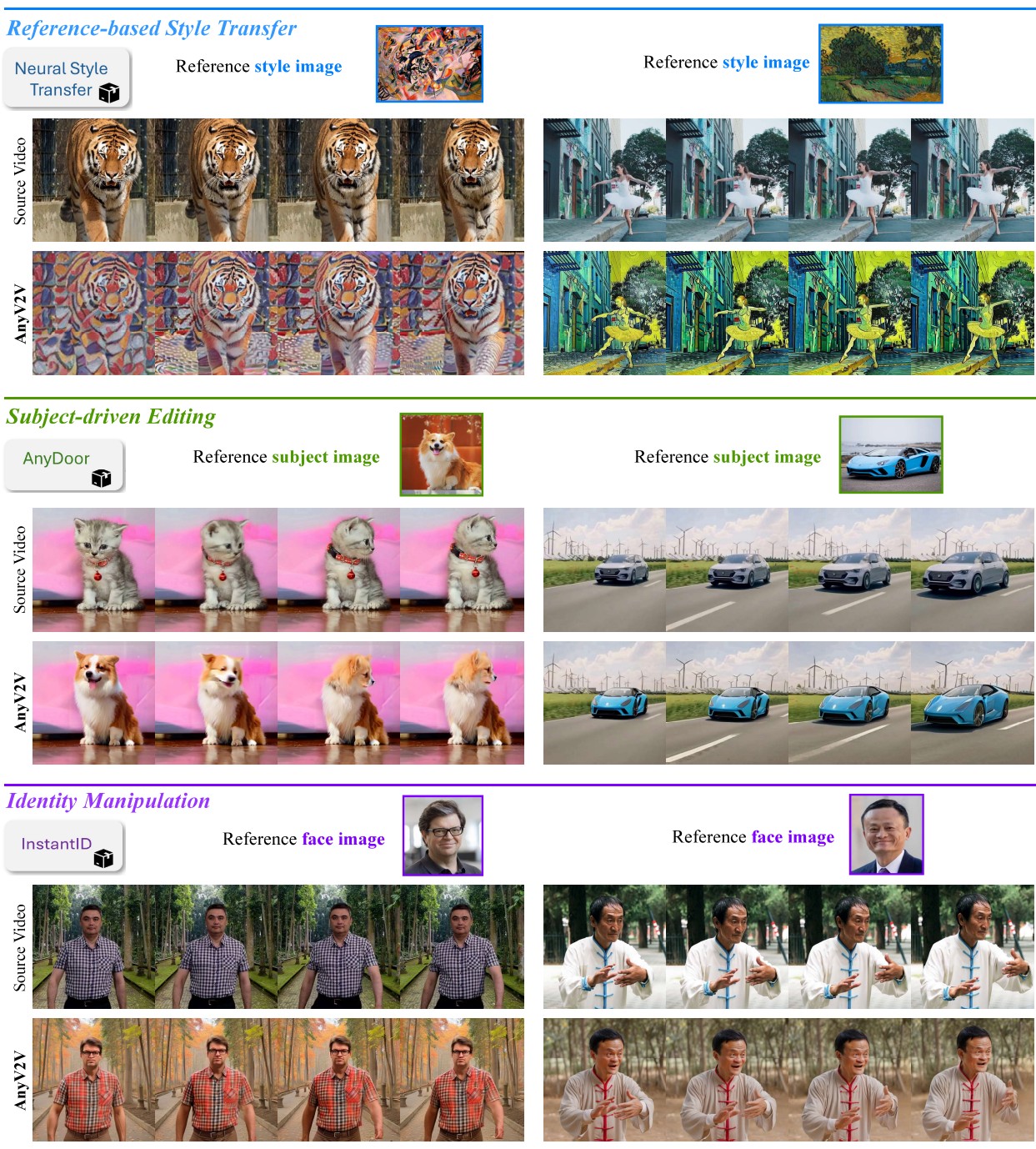

Figure 4: With different image editing models, AnyV2V can achieve a wide range of editing tasks, including reference-based style transfer, subject-driven editing, and identity manipulation.

2020) sampler and set $T$ to the default values of the selected I2V models. Following PnP (Tumanyan et al., 2023b), we set $l_1 = 4$ for convolution feature injection and $l_2 = l_3 = \{4, 5, 6, ..., 11\}$ for spatial and temporal attention injections. During sampling, we apply text classifier-free guidance (CFG) (Ho & Salimans, 2022) for all models with the same negative prompt *"Distorted, discontinuous, Ugly, blurry, low resolution, motionless, static, disfigured, disconnected limbs, Ugly faces, incomplete arms"* across all edits. To obtain the initial edited frames in our implementation, we use a set of image editing model candidates including prompt-based image editing model InstructPix2Pix (Brooks et al., 2023), style transfer model Neural Style Transfer (NST) (Gatys

Table 2: Quantitative comparisons for our AnyV2V with baselines on prompt-based video editing. Alignment: prompt alignment; Overall: overall preference. **Bold**: best results; Underline: top-2.

| Method | Human Evaluation ↑ | | CLIP Scores ↑ | |
|---|---|---|---|---|
| | Alignment | Overall | CLIP-Text | CLIP-Image |
| Tune-A-Video | 15.2% | 2.1% | 0.2902 | 0.9704 |
| TokenFlow | 31.7% | 20.7% | 0.2858 | **0.9783** |
| FLATTEN | 25.5% | 16.6% | 0.2742 | 0.9739 |
| AnyV2V (SEINE) | 28.9% | 8.3% | 0.2910 | 0.9631 |
| AnyV2V (ConsistI2V) | 33.8% | 11.7% | 0.2896 | 0.9556 |
| AnyV2V (I2VGen-XL) | **69.7%** | **46.2%** | **0.2932** | 0.9652 |

et al., 2015), subject-driven image editing model AnyDoor (Chen et al., 2023c), and identity-driven image editing model InstantID (Wang et al., 2024b). We experiment with only the successfully edited frames, which is crucial for our method. We conducted all the experiments on a single Nvidia A6000 GPU. To edit a 16-frame video, it requires around 15G GPU memory and around 100 seconds for the whole inference process. We refer readers to Appendix A for more discussions on our implementation details and hyperparameter settings.

## 5.2   Tasks Definition

1. **Prompt-based Editing:** allows users to manipulate video content using only natural language. This can include descriptive prompts or instructions. With the prompt, Users can perform a wide range of edits, such as incorporating accessories, spawning or swapping objects, adding effects, or altering the background.

2. **Reference-based Style Transfer:** In the realm of style transfer tasks, the artistic styles of Monet and Van Gogh are frequently explored, but in real-life examples, users might want to use a distinct style based on one particular artwork. In reference-based style transfer, we focus on using a style image as a reference to perform video editing. The edited video should capture the distinct style of the referenced artwork.

3. **Subject-driven Editing:** In subject-driven video editing, we aim at replacing an object in the video with a target subject based on a given subject image while maintaining the video motion and persevering the background.

4. **Identity Manipulation:** Identity manipulation allows the user to manipulate video content by replacing a person with another person's identity in the video based on an input image of the target person.

## 5.3   Evaluation Results

As shown in Figure 3 and 4, AnyV2V can perform the following video editing tasks: (1) prompt-based editing, (2) reference-based style transfer, (3) subject-driven editing, and (4) identity manipulation. we compare AnyV2V against three baseline models Tune-A-Video (Wu et al., 2023b), TokenFlow (Geyer et al., 2023) and FLATTEN (Cong et al., 2023) for (1) prompt-based editing. Since there exists no publicly available baseline method for task (2) (3) (4), we evaluate the performance of three I2V generation models under AnyV2V. We included a more comprehensive evaluation in Appendix B.

**Prompt-based Editing**   Unlike the baseline methods (Wu et al., 2023b; Geyer et al., 2023; Cong et al., 2023) that often introduce unwarranted changes not specified in the text commands, AnyV2V utilizes the precision of image editing models to ensure only the targeted areas of the scene are altered, leaving the rest unchanged. Combining AnyV2V with the instruction-guided image editing model InstructPix2Pix (Brooks et al., 2023), AnyV2V accurately places a party hat on an elderly man's head and correctly paints the plane

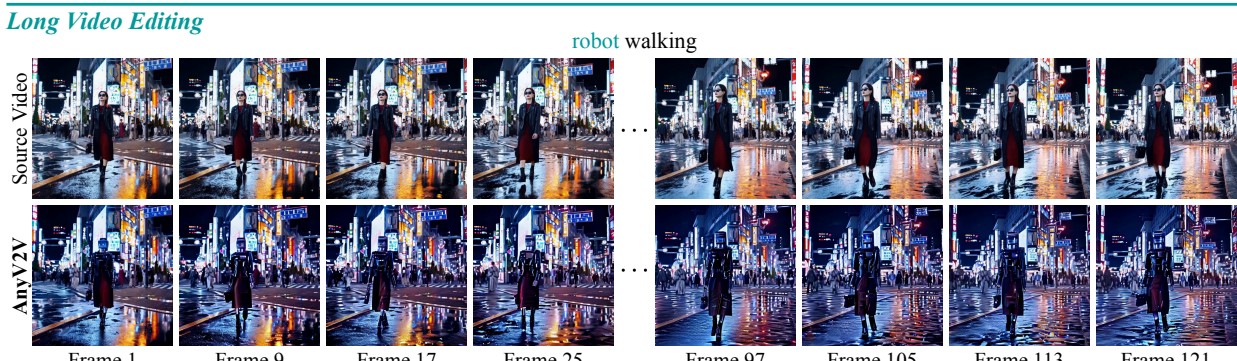

Figure 5: AnyV2V can edit video length beyond the training frame while maintaining motion consistency. The first row is the source video frames while the second rows are the edited. The editing prompt of the image was "turn woman into a robot" using image model InstructPix2Pix (Brooks et al., 2023).

in blue. Additionally, it maintains the original video's background and fidelity, whereas, in comparison, baseline methods often alter the color tone and shape of objects, as illustrated in Figure 3. Also, for motion tasks such as adding snowing weather, I2V models from AnyV2V provide inherent support for animating the snowing while the baseline methods would result in flickering. For quantitative evaluations, we conduct a human evaluation to examine the degree of prompt alignment and overall preference of the edited videos based on user voting, and also compute the metrics CLIP-Text for text alignment and CLIP-Image for temporal consistency. In detail, CLIP-Text is computed by the average cosine similarity between text embeddings from CLIP model (Radford et al., 2021) and CLIP-Image is computed in the same way but between image embeddings for every pair of consecutive frames. Table 2 shows that AnyV2V generally achieves high text-alignment and temporal consistency, while AnyV2V with I2VGen-XL backbone is the most preferred method because it does not edit the video precisely.

**Style Transfer, Subject-Driven Editing and Identity Manipulation**   For these novel tasks, we stress the alignment with the reference image instead of the text prompt. As shown in Figure 4, we can observe that for task (2), AnyV2V can capture one particular style that is tailor-made, and even if the style is not learned by the text encoder. In the examples, AnyV2V captures the style of Vassily Kandinsky's artwork "Composition VII" and Vincent Van Gogh's artwork "Chateau in Auvers at Sunset" accurately. For task (3), AnyV2V can replace the subject in the video with other subjects even if the new subject differs from the original subject. In the examples, a cat is replaced with a dog according to the reference image and maintains highly aligned motion and background as reflected in the source video. A car is replaced by our desired car while the wheel is still spinning in the edited video. For task (4), AnyV2V can swap a person's identity to anyone. We report both human evaluation results and find that AnyV2V with I2VGen-XL backbone is the most preferred method in terms of reference alignment and overall performance, plotted in Table 5, which is in Appendix B.

**I2V Backbones**   We find that AnyV2V (I2VGen-XL) tends to be the most robust both qualitatively and quantitatively. It has a good generalization ability to produce consistent motions in the video with high visual quality. AnyV2V (ConsistI2V) can generate consistent motion, but sometimes the watermark would appear due to its training data, thus harming the visual quality. AnyV2V (SEINE)'s generalization ability is relatively weaker but still produces consistent and high-quality video if the motion is simple enough, such as a person walking.

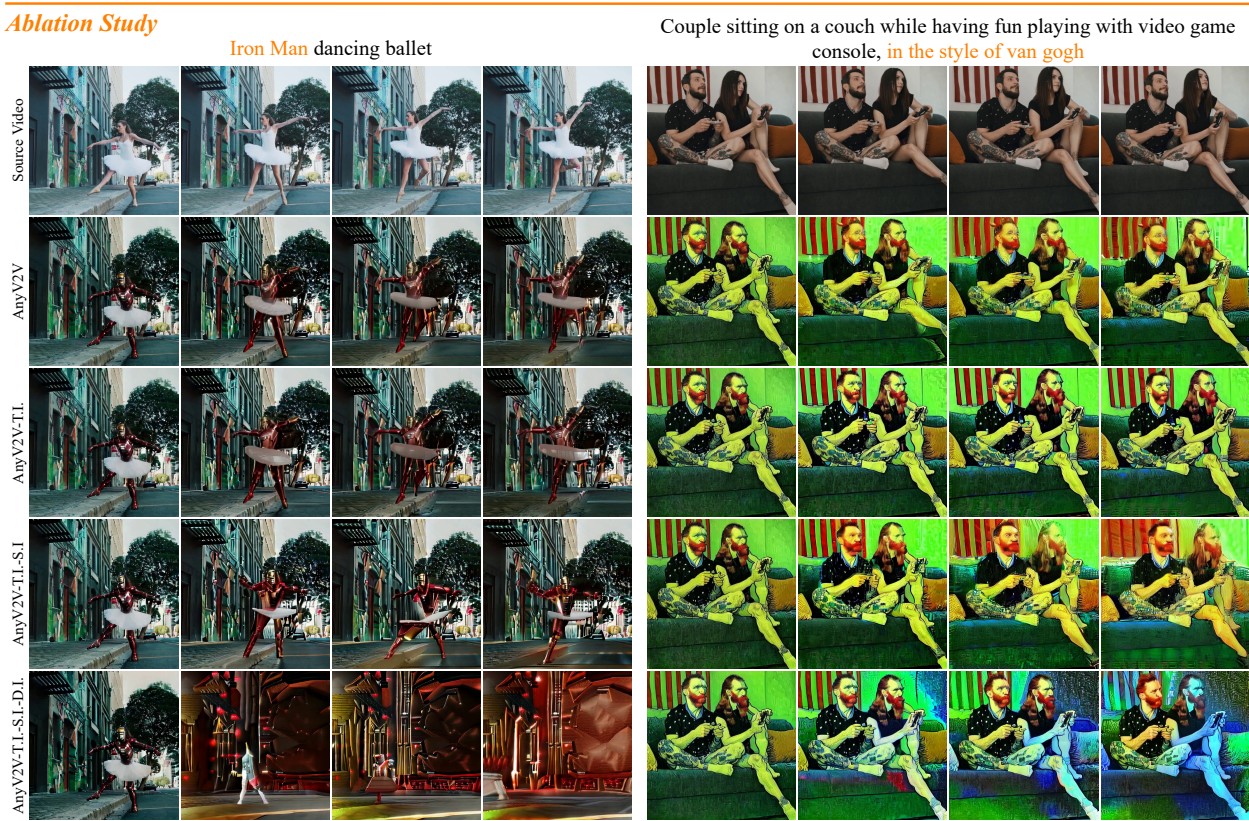

Figure 6: Visual comparisons of AnyV2V's editing results after disabling temporal feature injection (T.I.), spatial feature injection (S.I.) and DDIM inverted initial noise (D.I.).

**Editing Video beyond Training Frames of I2V model** Current state-of-the-art I2V models (Chen et al., 2023d; Ren et al., 2024; Zhang et al., 2023c) are mostly trained on video data that contains only 16 frames. To edit videos that have length beyond the training frames of the I2V model, an intuitive approach would be generating videos in an auto-regressive manner as used in ConsistI2V (Ren et al., 2024) and SEINE (Chen et al., 2023d). However, we find that such an experiment setup cannot maintain semantic consistency in our case. As many works in extending video generation exploit the initial latent to generate longer video (Qiu et al., 2023; Wu et al., 2023c), we leverage the longer inverted latent as the initial latent and force an I2V model to generate longer frames of output. Our experiments found that the inverted latent contains enough temporal and semantic information to allow the generated video to maintain temporal and semantic consistency, as shown in Figure 5.

## 5.4 Ablation Study

To verify the effectiveness of our design choices, we conduct an ablation study by iteratively disabling the three core components in our model: temporal feature injection, spatial feature injection, and DDIM inverted latent as initial noise. We use AnyV2V (I2VGen-XL) and a subset of 20 samples in this ablation study and report both the frame-wise consistency results using CLIP-Image score in Table 3 and qualitative comparisons in Figure 6. We provide more ablation analysis of other design considerations of our model in the Appendix.

**Effectiveness of Temporal Feature Injection** According to the results, after disabling temporal feature injection in AnyV2V (I2VGen-XL), while we observe a slight increase in the CLIP-Image score value, the edited videos often demonstrate less adherence to the motion presented in the source video. For example, in the second frame of the "couple sitting" case (3rd row, 2nd column in the right panel in Figure 6), the motion of the woman raising her leg in the source video is not reflected in the edited video without applying

Table 3: Ablation study results for AnyV2V (I2VGen-XL). T. Injection and S. Injection correspond to temporal and spatial feature injection mechanisms, respectively.

| Model | CLIP-Image ↑ |
|---|---|
| AnyV2V (I2VGen-XL) | 0.9648 |
| AnyV2V (I2VGen-XL) w/o T. Injection | 0.9652 |
| AnyV2V (I2VGen-XL) w/o T. Injection & S. Injection | 0.9637 |
| AnyV2V (I2VGen-XL) w/o T. Injection & S. Injection & DDIM Inversion | 0.9607 |

temporal injection. On the other hand, even when the style of the video is completely changed, AnyV2V (I2VGen-XL) with temporal injection is still able to capture this nuance motion in the edited video.

**Effectiveness of Spatial Feature Injection** As shown in Table 3, we observe a drop in the CLIP-Image score after removing the spatial feature injection mechanisms from our model, indicating that the edited videos are not smoothly progressed across consecutive frames and contain more appearance and motion inconsistencies. Further illustrated in the third row of Figure 6, removing spatial feature injection will often result in incorrect subject appearance and pose (as shown in the "ballet dancing" case) and degenerated background appearance (evident in the "couple sitting" case). These observations demonstrate that directly generating edited videos from the DDIM inverted noise is often not enough to fully preserve the source video structures, and the spatial feature injection mechanisms are crucial for achieving better editing results.

**DDIM Inverted Noise as Structural Guidance** Finally, we observe a further decrease in CLIP-Image scores and a significantly degraded visual appearance in both examples in Figure 6 after replacing the initial DDIM inverted noise with random noise during sampling. This indicates that the I2V generation models become less capable of animating the input image when the editing prompt is completely out-of-domain and highlights the importance of the DDIM inverted noise as the structural guidance of the edited videos.

## 6 Conclusion

In this paper, we presented AnyV2V, a novel unified framework for video editing. Our framework is training-free, highly cost-effective, and can be applied to any image editing model and I2V generation model. To perform video editing with high precision, we propose a two-stage approach to first edit the initial frame of the source video and then condition an I2V model with the edited first frame and the source video features and inverted latents to produce the edited video at any length. Comprehensive experiments have shown that our method achieves outstanding outcomes across a broad spectrum of applications that are beyond the scope of existing SOTA methods while achieving superior results on both common video metrics and human evaluation. For future work, we aim to find a tuning-free method to bridge the I2V properties into T2V models so that we can leverage existing strong T2V models for video editing.

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

# Appendix

## A    Discussion on Model Implementation Details

When adapting our AnyV2V to various I2V generation models, we identify two sets of hyperparameters that are crucial to the final video editing results. They are (1) selection of U-Net decoder layers ($l_1$, $l_2$ and $l_3$) to perform convolution, spatial attention and temporal attention injection and (2) Injection thresholds $\tau_{conv}$, $\tau_{sa}$ and $\tau_{ta}$ that control feature injections to happen in specific diffusion steps. In this section, we provide more discussions and analysis on the selection of these hyperparameters.

### A.1    U-Net Layers for Feature Injection

To better understand how different layers in the I2V denoising U-Net produce features during video sampling, we perform a visualization of the convolution, spatial and temporal attention features for the three candidate I2V models I2VGen-XL (Zhang et al., 2023c), ConsistI2V (Ren et al., 2024) and SEINE (Chen et al., 2023d). Specifically, we visualize the average activation values across all channels in the output feature map from the convolution layers, and the average attention scores across all attention heads and all tokens (i.e. average attention weights for all other tokens attending to the current token). The results are shown in Figure 7.

According to the figure, we observe that the intermediate convolution features from different I2V models show similar characteristics during video generation: earlier layers in the U-Net decoder produce features that represent the overall layout of the video frames and deeper layers capture the high-frequency details such as edges and textures. We choose to set $l_1 = 4$ for convolution feature injection to inject background and layout guidance to the edited video without introducing too many high-frequency details. For spatial and temporal attention scores, we observe that the spatial attention maps tend to represent the semantic regions in the video frames while the temporal attention maps highlight the foreground moving subjects (e.g. the running woman in Figure 7). One interesting observation for I2VGen-XL is that its spatial attention operations in deeper layers almost become hard attention, as the spatial tokens only attend to a single or very few tokens in each frame. We propose to inject features in decoder layers 4 to 11 ($l_2 = l_3 = \{4, 5, ..., 11\}$) to preserve the semantic and motion information from the source video.

### A.2    Ablation Analysis on Feature Injection Thresholds

We perform additional ablation analysis using different feature injection thresholds to study how these hyperparameters affect the edited video.

**Effect of Spatial Injection Thresholds** $\tau_{conv}, \tau_{sa}$    We study the effect of disabling spatial feature injection or using different $\tau_{conv}$ and $\tau_{sa}$ values during video editing and show the qualitative results in Figure 8. We find that when spatial feature injection is disabled, the edited videos fail to fully adhere to the layout and motion from the source video. When spatial feature injection thresholds are too high, the edited videos are corrupted by the high-frequency details from the source video (e.g. textures from the woman's down jacket in Figure 8). Setting $\tau_{conv} = \tau_{sa} = 0.2T$ achieves a desired editing outcome for our experiments.

**Effect of Temporal Injection Threshold** $\tau_{ta}$    We study the hyperparameter of temporal feature injection threshold $\tau_{ta}$ in different settings and show the results in Figure 9. We observe that in circumstances where $\tau_{ta} < 0.5T$ ($T$ is the total denoising steps), the motion guidance is too weak that it leads to only partly aligned motion with the source video, even though the motion itself is logical and smooth. At $\tau_{ta} > 0.5T$, the generated video shows a stronger adherence to the motion but distortion occurs. We employ $\tau_{ta} = 0.5T$ in our experiments and find that this value strikes the perfect balance on motion alignment, motion consistency, and video fidelity.

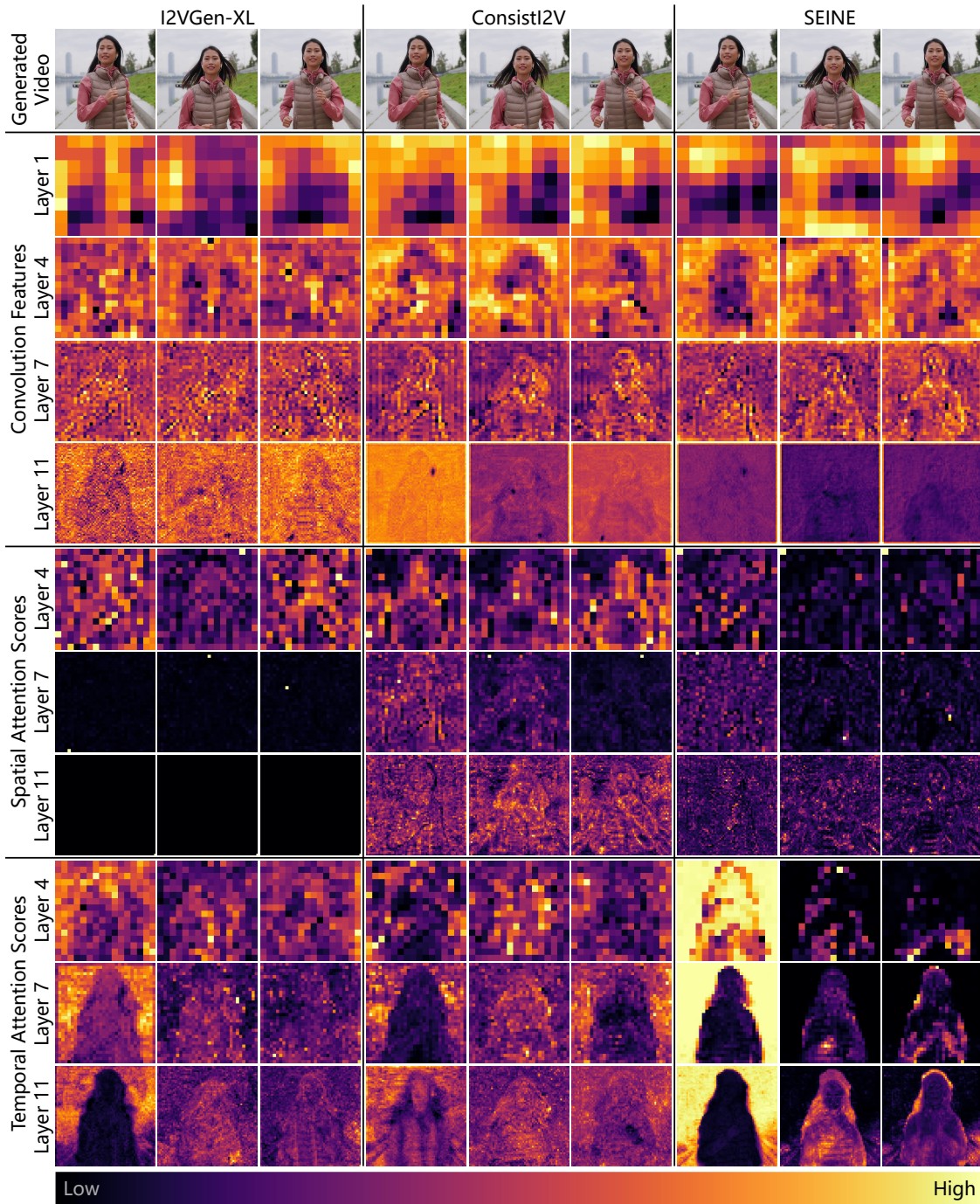

Figure 7: Visualizations of the convolution, spatial attention and temporal attention features during video sampling for I2V generation models' decoder layers. We feed in the DDIM inverted noise to the I2V models such that the generated videos (first row) are reconstructions of the source video.

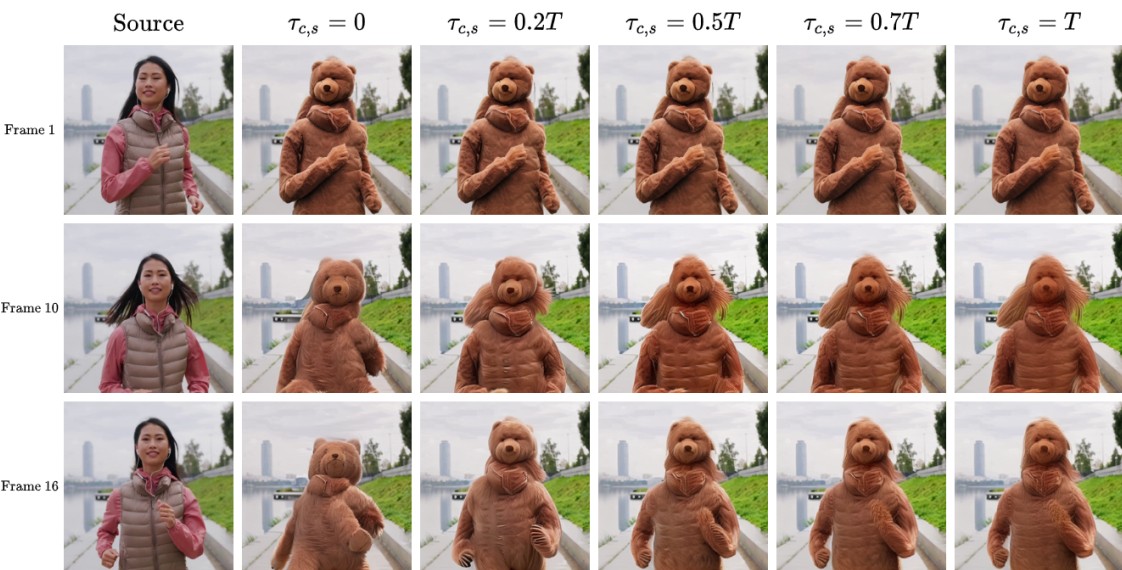

Figure 8: Hyperparameter study on spatial feature injection. We find that $\tau_{sa} = 0.2T$ is the best setting for maintaining the layout and structure in the edited video while not introducing unnecessary visual details from the source video. $\tau_{c,s}$ represents $\tau_{conv}$ and $\tau_{sa}$. (Editing prompt: *teddy bear running*. The experiment was conducted with the I2VGen-XL backbone.

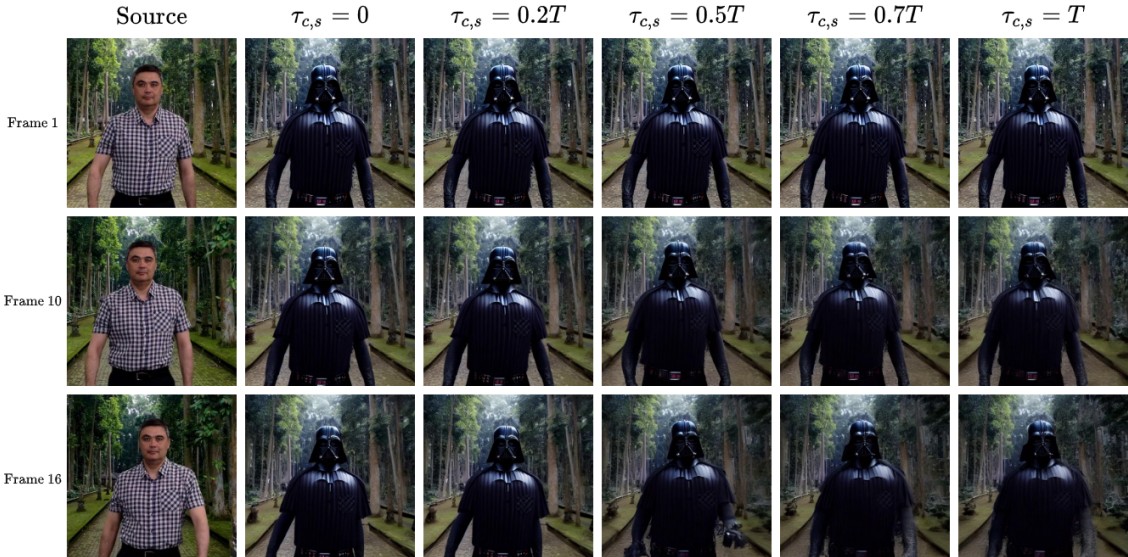

Figure 9: Hyperparameter study on temporal feature injection. We find that $\tau_{ta} = 0.5T$ to be the optimal setting as it balances motion alignment, motion consistency, and video fidelity. (Editing prompt: *darth vader walking*. The experiment was conducted with the SEINE backbone.

# B  Evaluation Detail

## B.1  Quantitative Evaluations

**Prompt-based Editing**  For (1) prompt-based editing, we conduct a human evaluation to examine the degree of prompt alignment and overall preference of the edited videos based on user voting. We compare

AnyV2V against three baseline models: Tune-A-Video (Wu et al., 2023b), TokenFlow (Geyer et al., 2023) and FLATTEN (Cong et al., 2023). Human evaluation results in Table 2 demonstrate that our model achieves the best overall preference and prompt alignment among all methods, and AnyV2V (I2VGen-XL) is the most preferred method. We conjecture that the gain is coming from our compatibility with state-of-the-art image editing models.

We also employ automatic evaluation metrics on our edited video of the human evaluation datasets. Following previous works (Ceylan et al., 2023; Bai et al., 2024), our automatic evaluation employs the CLIP (Radford et al., 2021) model to assess both text alignment and temporal consistency. For text alignment, we calculate the CLIP-Score, specifically by determining the average cosine similarity between the CLIP text embeddings derived from the editing prompt and the CLIP image embeddings across all frames. For temporal consistency, we evaluate the average cosine similarity between the CLIP image embeddings of every pair of consecutive frames. These two metrics are referred to as CLIP-Text and CLIP-Image, respectively. Our automatic evaluations in Table 2 demonstrate that our model is competitive in prompt-based editing compared to baseline methods.

**Reference-based Style Transfer; Identity Manipulation and Subject-driven Editing**   For novel tasks (2), (3) and (4), we evaluate the performance of three I2V generation models using human evaluations and show the results in Table 5. As these tasks require reference images instead of text prompts, we focus on evaluating the reference alignment and overall preference of the edited videos. According to the results, we observe that AnyV2V (I2VGen-XL) is the best model across all tasks, underscoring its robustness and versatility in handling diverse video editing tasks. AnyV2V (SEINE) and AnyV2V (ConsistI2V) show varied performance across tasks. AnyV2V (SEINE) performs good reference alignment in reference-based style transfer and identity manipulation, but falls short in subject-driven editing with lower scores. On the other hand, AnyV2V (ConsistI2V) shines in subject-driven editing, achieving second-best results in both reference alignment and overall preference. Since the latest image editing models have not yet reached a level of maturity that allows for consistent and precise editing (Ku et al., 2024), we also report the image editing success rate in Table 5 to clarify that our method relies on a good image frame edit.

## B.2   Qualitative Results

**Prompt-based Editing**   By leveraging the strength of image editing models, our AnyV2V framework provides precise control of the edits such that the irrelevant parts in the scene are untouched after editing. In our experiment, we used InstructPix2Pix (Brooks et al., 2023) for the first frame edit. Shown in Figure 3, our method correctly places a party hat on an old man's head and successfully turns the color of an airplane to blue, while preserving the background and keeping the fidelity to the source video. Comparing our work with the three baseline models TokenFlow (Geyer et al., 2023), FLATTEN (Cong et al., 2023), and Tune-A-Video (Wu et al., 2023b), the baseline methods display either excessive or insufficient changes in the edited video to align with the editing text prompt. The color tone and object shapes are also tilted. It is also worth mentioning that our approach is far more consistent on some motion tasks such as adding snowing weather, due to the I2V model's inherent support for animating still scenes. The baseline methods, on the other hand, can add snow to individual frames but cannot generate the effect of snow falling, as the per-frame or one-shot editing methods lack the ability of temporal modeling.

**Reference-based Style Transfer**   Our approach diverges from relying solely on textual descriptors for conducting style edits, using the style transfer model NST (Gatys et al., 2015) to obtain the edited frame. This level of controllability offers artists the unprecedented opportunity to use their art as a reference for video editing, opening new avenues for creative expression. As demonstrated in Figure 4, our method captures the distinctive style of Vassily Kandinsky's artwork "Composition VII" and Vincent Van Gogh's artwork "Chateau in Auvers at Sunset" accurately, while such an edit is often hard to perform using existing text-guided video editing methods.

**Subject-driven Editing**   In our experiment, we employed a subject-driven image editing model Any-Door (Chen et al., 2023c) for the first frame editing. AnyDoor allows replacing any object in the target image with the subject from only one reference image. We observe from Figure 4 that AnyV2V produces highly

motion-consistent videos when performing subject-driven object swapping. In the first example, AnyV2V successfully replaces the cat with a dog according to the reference image and maintains highly aligned motion and background as reflected in the source video. In the second example, the car is replaced by our desired car while maintaining the rotation angle in the edited video.

**Identity Manipulation** By integrating the identity-preserved image personalization model InstantID (Wang et al., 2024b) with ControlNet (Zhang et al., 2023b), this approach enables the replacement of an individual's identity to create an initial frame. Our AnyV2V framework then processes this initial frame to produce an edited video, swapping the person's identity as showcased in Figure 4. To the best of our knowledge, our work is the first to provide such flexibility in the video editing models. Note that the InstantID with ControlNet method will alter the background due to its model property. It is possible to leverage other identity-preserved image personalization models and apply them to AnyV2V to preserve the background.

### B.3 Human Evaluation

#### B.3.1 Dataset

Our human evaluation dataset contains a total of 89 samples that have been collected from `https://www.pexels.com`. For prompt-based editing, we employed InstructPix2Pix (Brooks et al., 2023) to compose the examples. Topics include swapping objects, adding objects, and removing objects. For subject-driven editing, we employed AnyDoor (Chen et al., 2023c) to replace objects with reference subjects. For Neural Style Transfer, we employed NST (Gatys et al., 2015) to compose the examples. For identity manipulation, we employed InstantID (Wang et al., 2024b) to compose the examples. See Table 4 for the statistic.

Table 4: Number of entries for Video Editing Evaluation Dataset

| Category | Number of Entries | Image Editing Models Used |
|---|---|---|
| Prompt-based Editing | 45 | InstructPix2Pix |
| Reference-based Style Transfer | 20 | NST |
| Identity Manipulation | 13 | InstantID |
| Subject-driven Editing | 11 | AnyDoor |
| Total | 89 | |

#### B.3.2 Interface

In the evaluation setup, we provide the evaluator with generated images from both the baseline models and the AnyV2V models. Evaluators are tasked with selecting videos that best align with the provided prompt or the reference image. Additionally, they are asked to express their overall preference for the edited videos. For a detailed view of the interface used in this process, please see Figure 10.

## C Discussion

### C.1 Limitations

**Inaccurate Edit from Image Editing Models.** As our method relies on an initial frame edit, the image editing models are used. However, the current state-of-the-art models are not mature enough to perform accurate edits consistently (Ku et al., 2024). For example, in the subject-driven video editing task, we found that AnyDoor (Chen et al., 2023c) requires several tries to get a good editing result. Efforts are required in manually picking a good edited frame. We expect that in the future better image editing models will minimize such effort.

**Limited ability of I2V models.** We found that the results from our method cannot follow the source video motion if the motion is fast (e.g. billiard balls hitting each other at full speed) or complex (e.g. a

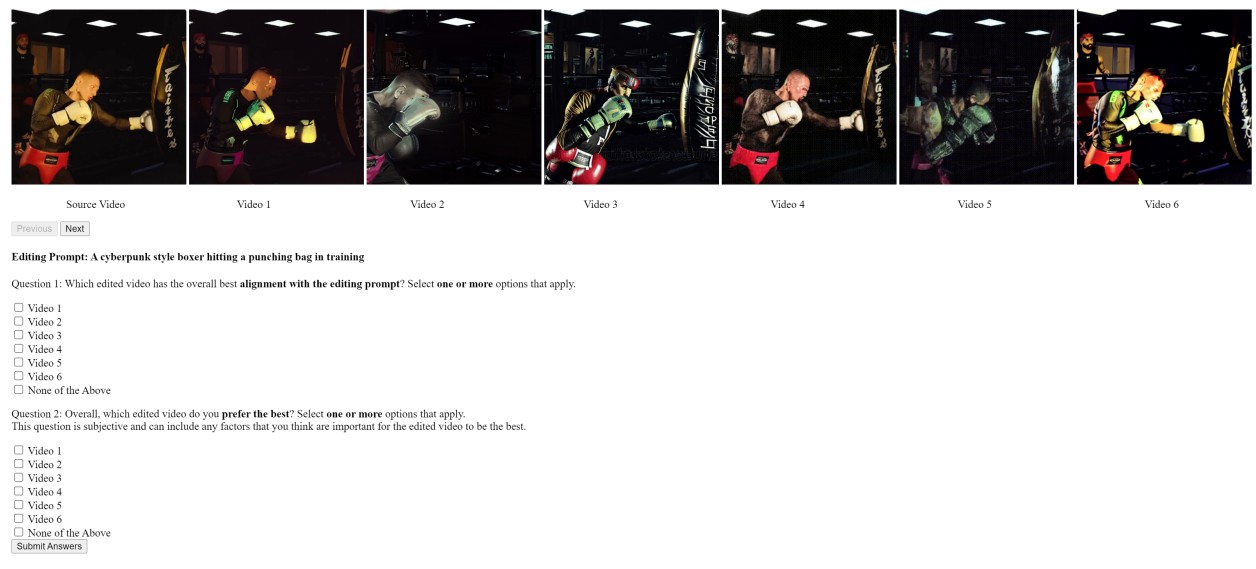

Figure 10: The interface of individual evaluation.

Table 5: Comparisons for three I2V models under AnyV2V framework on novel video editing tasks. Align: reference alignment; Overall: overall preference. **Bold**: best results; Underline: top-2.

| Task | Reference-based Style Transfer | | Subject-driven Editing | | Identity Manipulation | |
|---|---|---|---|---|---|---|
| Image Editing Method | NST | | AnyDoor | | InstantID | |
| Image Editing Success Rate | ≈90% | | ≈10% | | ≈80% | |
| Human Evaluation | Align ↑ | Overall ↑ | Align ↑ | Overall ↑ | Align ↑ | Overall ↑ |
| AnyV2V (SEINE) | 92.3% | 30.8% | 48.4% | 15.2% | 72.7% | 18.2% |
| AnyV2V (ConsistI2V) | 38.4% | 10.3% | 63.6% | 42.4% | 72.7% | 27.3% |
| AnyV2V (I2VGen-XL) | **100.0%** | **76.9%** | **93.9%** | **84.8%** | **90.1%** | **45.4%** |

person clipping her hair). One possible reason is that the current popular I2V models are generally trained on slow-motion videos, such that lacking the ability to regenerate fast or complex motion even with motion guidance. We anticipate that the presence of a robust I2V model can address this issue.

## C.2 License of Assets

For image editing models, InstructPix2Pix (Brooks et al., 2023) inherits Creative ML OpenRAIL-M License as it is built upon Stable Diffusion. Neural Style Transfer (Gatys et al., 2015) is under Creative Commons Attribution 4.0 License, and code samples are licensed under the Apache 2.0 License. InstantID (Wang et al., 2024b) is under Apache License 2.0. AnyDoor (Chen et al., 2023c) is under the MIT License.

For baselines, Tune-A-Video (Wu et al., 2023b) is under Apache License 2.0, TokenFlow (Geyer et al., 2023) is under MIT License andFLATTEN (Cong et al., 2023) is under Apache License 2.0.

For the human evaluation dataset, the dataset has been collected from `https://www.pexels.com`, with all data governed by the terms outlined at `https://www.pexels.com/license/`.

We decide to release AnyV2V code under the Creative Commons Attribution 4.0 License for easy access in the research community.

### C.3 Societal Impacts

**Postive Social Impact.** AnyV2V has the potential to significantly enhance the capabilities of video editing systems, making it easier for a wider range of users to manipulate images. This could have numerous positive social impacts, as users would be able to achieve their editing goals without needing professional editing knowledge, such as using Photoshop or painting.

**Misinformation spread and Privacy violations.** As our technique allows for object manipulation, it can produce highly realistic yet completely fabricated videos of one individual or subject. There is a risk that harmful actors could exploit our system to generate counterfeit videos to disseminate false information. Moreover, the ability to create convincing counterfeit content featuring individuals without their permission undermines privacy protections, possibly leading to the illicit use of a person's likeness for harmful purposes and damaging their reputation. These issues are similarly present in DeepFake technologies. To mitigate the risk of misuse, one proposed solution is the adoption of unseen watermarking, a method commonly used to tackle such concerns in image generation.

### C.4 Safeguards

It is crucial to implement proper safeguards and responsible AI frameworks when developing user-friendly video editing systems. For the human evaluation dataset, we manually collect a diverse range of images to ensure a balanced representation of objects from various domains. We only collect images that are considered safe.

### C.5 Ethical Concerns for Human Evaluation

We believe our proposed human evaluation does not incur ethical concerns due to the following reasons: (1) the study does not involve any form of intervention or interaction that could affect the participants' well-being. (2) Rating video content does not involve any physical or psychological risk, nor does it expose participants to sensitive or distressing material. (3) The data collected from participants will be entirely anonymous and will not contain any identifiable private information. (4) Participation in the study is entirely voluntary, and participants can withdraw at any time without any consequences.

### C.6 New Assets

Our paper introduces several new assets including a human evaluation dataset and demo videos generated by AnyV2V. Each asset is thoroughly documented, detailing its creation, usage, and any relevant methodologies. The human evaluation dataset documentation includes details on how participant consent was obtained. The demo videos are provided as an anonymized zip file to comply with submission guidelines, with detailed instructions for use. All assets are shared under an open license to facilitate reuse and further research.

## D    More AnyV2V Showcases

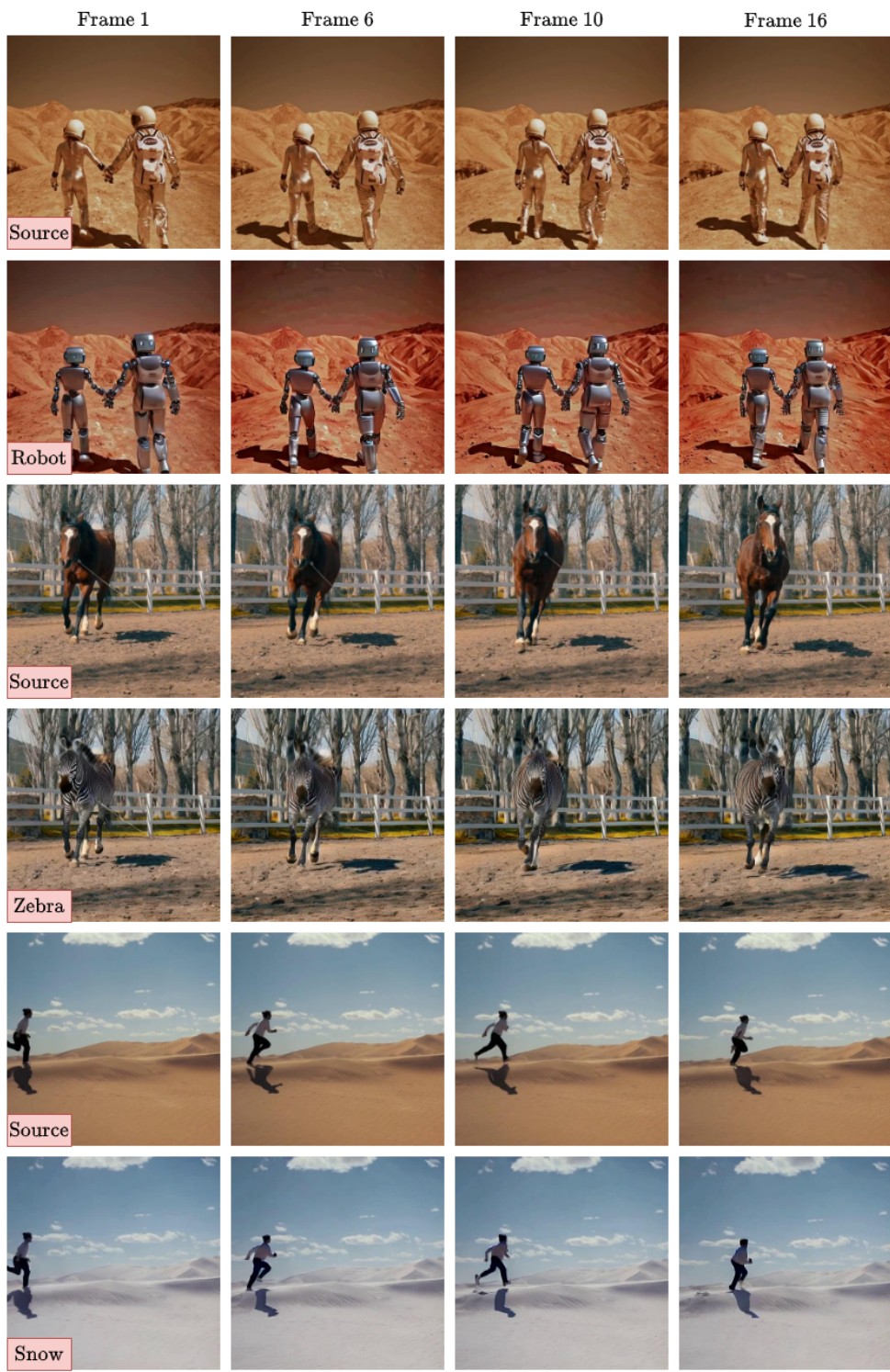

Figure 11: AnyV2V becomes an instruction-based video editing tool when plugged with instruction-guided image editing models like InstructPix2Pix (Brooks et al., 2023). Prompt used "Turn the couple into robots", "Turn horse into zebra", and "Turn the sand into snow".

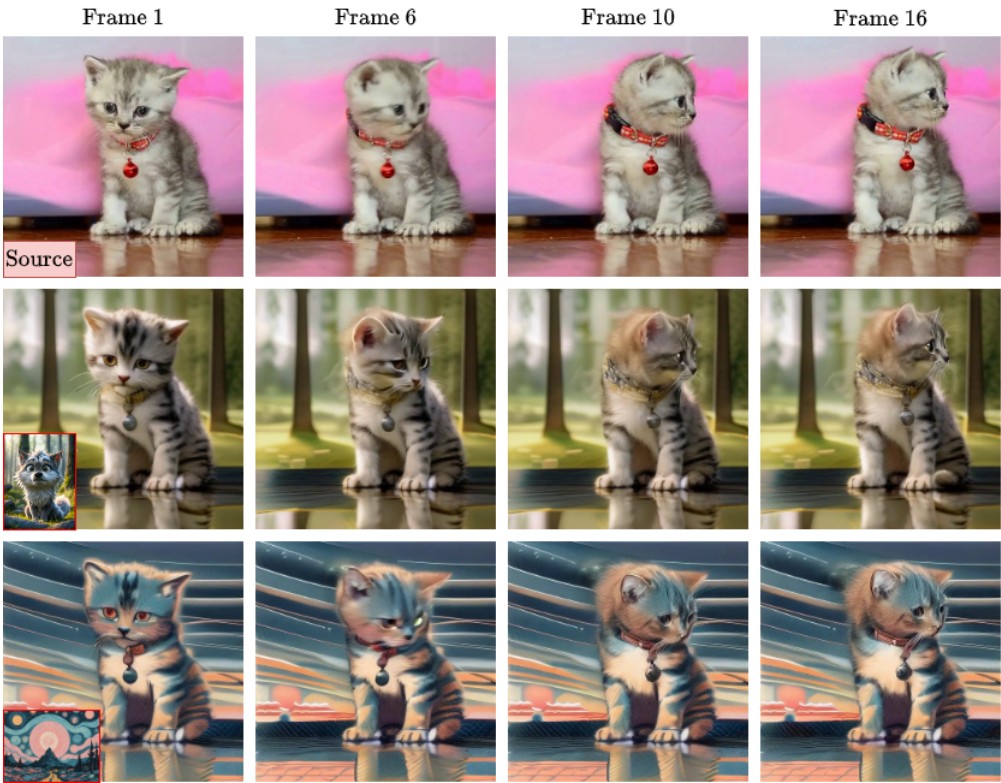

Figure 12: The more recent model InstantStyle (Wang et al., 2024a) can seamlessly plug in AnyV2V to perform reference-based style transfer video editing. The reference art image in the bottom left corner is used to retrieve the first image edit.

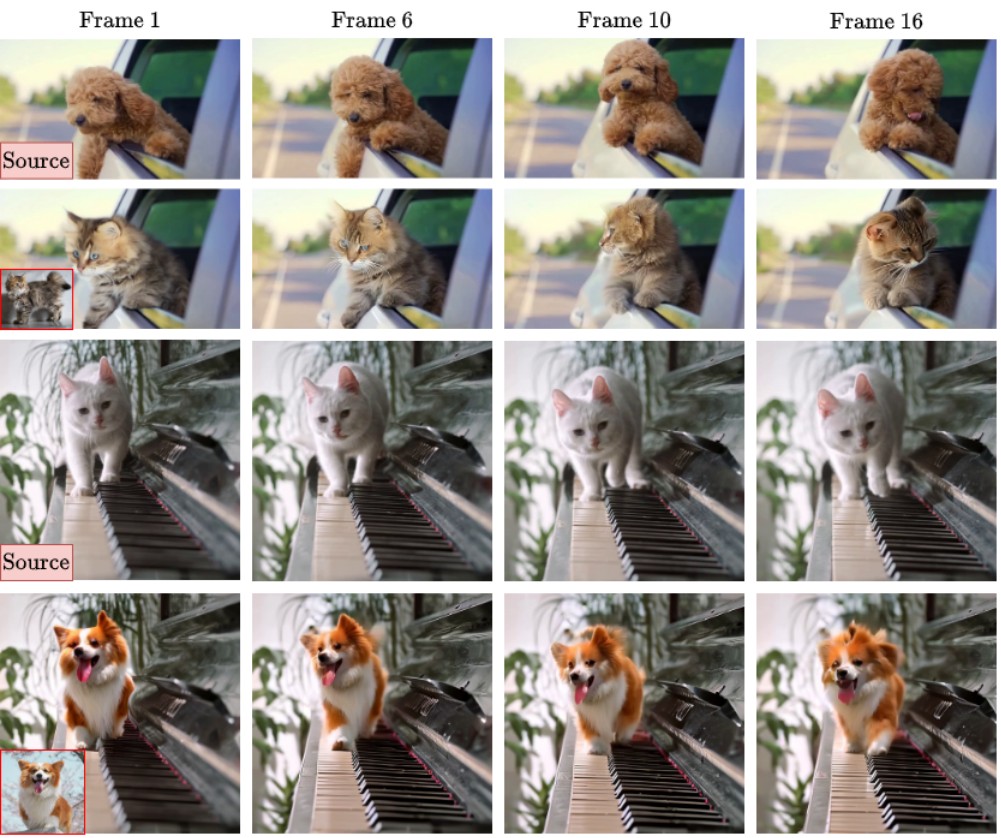

Figure 13: AnyV2V can perform subject-driven video editing with a single image reference, using a subject-driven image editing model like AnyDoor (Chen et al., 2023c). The subject image in the bottom left corner is used to retrieve the first image edit.

