# OpenReview forum: "AnyV2V: A Tuning-Free Framework For Any Video-to-Video Editing Tasks"
_TMLR — Accepted by TMLR_

### Review · Reviewer_VExa · 2024-08-23

**Summary Of Contributions:**

This manuscripts proposed a tuning-free framework for video to video transformation, leveraging different kinds of existing pre-trained image editing model and image to video model. The core idea is first editing the first frame, then generating the video with the structural information as well as the spatial and temporal information obtained from the DDIM inversion.

**Audience:**

Yes

**Broader Impact Concerns:**

Not applicable.

**Claims And Evidence:**

Yes

**Requested Changes:**

Please elaborate more on how the temporal information is leveraged during the decoding. Also please separate the diffusion steps and frame steps to make the presentation clear.

**Strengths And Weaknesses:**

### Strengths:
* The idea is surprisingly simple and it does not require further fine-tuning on existing model.

***
### Weaknesses:
* The methodology parts is too abbreviating. It’s unclear how the frames other than the first frame is encoded into the latent and help the video editing.
* From a high-level perspective, this paper can be viewed as an extension of the PnP diffusion features in video domain, by additionally injecting the temporal information into the decoding model. However, as I mentioned, this part is not presented clearly.

---

> ### Author Response · Authors · 2024-08-27
> **Response**
>
> We are glad that you perceive our work’s simplicity as a strength. We here address your concerns correspondingly:
>
> > W1: The methodology parts is too abbreviating. It’s unclear how the frames other than the first frame is encoded into the latent and help the video editing.
>
> Response to W1: In Section 4.2, we explained how the frames other than the first frame are encoded into the latent. We extracted the information from the frames of source video, using DDIM inversion. Inversion in diffusion models is done by reversing the diffusion scheduler timesteps, thus this inverted latent contains the information of the source video, providing structural guidance for the video editing.
>
> > W2: From a high-level perspective, this paper can be viewed as an extension of the PnP diffusion features in video domain, by additionally injecting the temporal information into the decoding model. However, as I mentioned, this part is not presented clearly.
>
> Response to W2: This paper did leverages PnP diffusion features and extends them into temporal features. However, another important component of this work is to leverage the generalization ability of I2V models, allowing first-frame control. This paradigm linked the image editing and video editing problems together. This opened many potential task applications and our work has demonstrated tasks like subject-driven video editing, identity manipulation, and referenced style video transfer that cannot be done by previous works (shown in Table 1). Regarding the concern of unclear presentation in temporal feature injection, we recommend checking Appendix A. We studied the visualization of temporal attention features in Appendix A.1 and also conducted the Ablation Analysis on Feature Injection Thresholds in Appendix A.2, which contains the effect according to different injection diffusion steps and frame steps. We demonstrated that temporal feature injection helps guide the motions in video editing.

---

### Review · Reviewer_3TgA · 2024-08-24

**Summary Of Contributions:**

This paper proposed to a video editing method by combining existing image editing models with image-to-video (I2V) models. The main idea is to apply the existing, desired image editing models on the first frame, and then utilize the information extracted from the source video, including temporal and spatial features and DDIM inverted latent to create video with proper injections. Because all the editing and I2V models are found in previous works, no additional training or tuning is needed. Four types of editing approaches combined with three I2V models were demonstrated in this study. (not all the 4x3 combinations though.) Quantitative comparisons with previous works, using human evaluation scores and CLIP scores, show that the proposed method is superior in human evaluations and comparable in CLIP scores.

**Audience:**

Yes

**Broader Impact Concerns:**

author already include this part in appendix C.3.

**Claims And Evidence:**

No

**Requested Changes:**

please see weakness above

**Strengths And Weaknesses:**

Strengths:
1. sufficient experimental data to demonstrate the flexibility and the quality of the proposed method.
2. relatively simple and straightforward approach that does not require additional model training/tuning.
3. open sourced

Weaknesses:
1. CLIP-image score does not seem to be a good metrics in this work, as author discussed in Section 5.4 - Temporal Feature Injection, i.e. while this injection is disabled and CLIP-score increased, author still believes the quality of the video is degraded. Similarly in Table 2, the three baseline models outperformed the proposed methods in CLIP-image scores but contradicted with human evaluation greatly. Including inconsistent metrics might cause confusions to the readers.

2. The author refers the proposed method as an image editing framework mainly because the information is extracted from the source and injected into the edited video in a frame by frame fashion. However, if the goal is not to generate new content but only editing existing frames, some of the applications shown in this work could simply be achieved by applying image editing on every single frame (without invoking I2V and injections), e.g. style transfer and change the color of an object. Maybe the author can add a few sentences to elaborate the pros and cons vs this 'naive approach'.

3. In abstract, the last sentence claims that the proposed method "significantly outperforms other baseline methods in automatic and human evaluations by a significant margin..." However, results in Table 2 do not seem to support this claim. While Table 2 does show that Human Evaluations are significantly better than prior arts, CLIP scores are comparable or even slightly worse in some cases. Author may want to provide additional data to support, or consider revise, this claim.

---

> ### Author Response · Authors · 2024-08-28
> **Response**
>
> Thanks for your valuable comments and constructive suggestions. We have put our response to your concerns one by one in the following:
>
> > W1: CLIP-image score does not seem to be a good metrics in this work, as author discussed in Section 5.4 - Temporal Feature Injection, i.e. while this injection is disabled and CLIP-score increased, author still believes the quality of the video is degraded. Similarly in Table 2, the three baseline models outperformed the proposed methods in CLIP-image scores but contradicted with human evaluation greatly. Including inconsistent metrics might cause confusions to the readers.
>
> Response to W1: We are aware that CLIP-based metrics are not very reliable and their lack of reliability has been mentioned in a few pieces of literature. Yet many related works still included such metrics. Thus we still want to include CLIP-based metrics in our work. We would like to clarify that CLIP-image scores focus more on the temporal consistency between different frames in the edited video and therefore do not reflect the overall quality of the edited videos. During our experiments, we observed that in some cases other methods will completely fail at making the desired edit based on the editing prompt, resulting in the editing video being almost identical to the source video. In such cases, the CLIP-image scores of these methods will be very high, as the output video frames are still consistent and smooth. However, they are likely to receive low scores in human evaluations due to the inability to follow editing prompts. Therefore, we consider human evaluation as the primary metric for evaluating video editing performances.
>
> > W2: The author refers the proposed method as an image editing framework mainly because the information is extracted from the source and injected into the edited video in a frame by frame fashion. However, if the goal is not to generate new content but only editing existing frames, some of the applications shown in this work could simply be achieved by applying image editing on every single frame (without invoking I2V and injections), e.g. style transfer and change the color of an object. Maybe the author can add a few sentences to elaborate the pros and cons vs this 'naive approach'.
>
> Response to W2: First, we would like to clarify that our method does not edit videos in a frame-by-frame fashion. In contrast, we use an image-to-video generation model to jointly edit and render all the frames in the video, guided by the edited first frame. An important advantage of our method is that during this joint editing process, the pre-trained temporal attention layers in the I2V models allow frames to attend to each other and exchange information with other frames. This enforces the smooth transitions between the edited frames and ensures the coherency of the whole video. On the other hand, simply applying image editing methods to individual frames will result in highly inconsistent content across different frames as the temporal features are neglected. Such temporal inconsistency has also been discussed in previous works such as in Section 5.1 of TokenFlow [1]. Here, we also show an example comparison between our method and simple per-frame image editing (the editing prompt of this example is "a rainbow colored dog running"):
>
> Source video: https://i.ibb.co/grz7MCg/input-fps301-ezgif-com-video-to-gif-converter.gif
>
> Our result: https://i.ibb.co/HgsBd7y/edited-video020205-ezgif-com-video-to-gif-converter.gif
>
> Per-frame image editing result: https://i.ibb.co/WPQwPz2/pnp-per-frame-baseline-fps-301-ezgif-com-video-to-gif-converter.gif
>
> As shown in the example, it is impossible to maintain temporal consistency for the per-frame image editing method, whereas our method successfully performs the desired edit and produces a smooth video at the same time. In the reviewer's proposed “naive approach” it would indeed be sufficient for some easy tasks (e.g. changing the color of an object). However, in most applications, applying image editing on every single frame would result in highly inconsistent content across frames because the temporal features are neglected, making it impossible for tasks like style transfer or object transformations.

---

> > ### Author Response · Authors · 2024-08-28
> > **Response (Cont'd)**
> >
> > > W3: In abstract, the last sentence claims that the proposed method "significantly outperforms other baseline methods in automatic and human evaluations by a significant margin..." However, results in Table 2 do not seem to support this claim. While Table 2 does show that Human Evaluations are significantly better than prior arts, CLIP scores are comparable or even slightly worse in some cases. Author may want to provide additional data to support, or consider revise, this claim.
> >
> > Response to W3: We will revise the claim to “significantly outperform other baseline methods in human evaluations and achieve comparable CLIP scores”. It is true that only human evaluations are significantly better than previous works but CLIP scores are comparable.
> >
> > [1] Geyer, Michal, et al. "Tokenflow: Consistent diffusion features for consistent video editing." arXiv preprint arXiv:2307.10373 (2023).

---

> > > ### Comment · Reviewer_3TgA · 2024-09-03
> > >
> > > Thanks for the explanations and responses from the authors. Based on the third comment, I would like to change the 'Claims And Evidence' item in my evaluation from 'No' to 'Yes'.

---

### Review · Reviewer_oTJF · 2024-09-17

**Summary Of Contributions:**

The introduction of AnyV2V as a novel tuning-free framework is a notable advancement in video editing research.

**Audience:**

Yes

**Claims And Evidence:**

Yes

**Requested Changes:**

Can you provide some case studies? In the four tasks mentioned in the experiment, which tasks did the AnyV2V model perform poorly? And in which scenarios did the generated videos perform poorly in these four tasks?

In addition to the prompt-based video editing task in Tab. 2, can you compare AnyV2V, which is based on the I2V model, with other video editing models that are based on the T2I model, such as CCEdit: Creative and Controllable Video Editing via Diffusion Models, presented at CVPR 2024, in the other three tasks?

Can you provide more visual results of AnyV2V in video editing? For example, in the form of videos. Also, some visual comparison results with other video editing methods would help demonstrate the performance of AnyV2V in video editing.

**Strengths And Weaknesses:**

The framework's capability to support various video editing tasks, including prompt-based editing, reference-based style transfer, subject-driven editing, and identity manipulation, demonstrates its broad applicability.

The article presents experiments showing that AnyV2V significantly outperforms other baseline methods in both automatic and human evaluations, which is a testament to its effectiveness. By breaking down the video editing task into a single image editing and image-to-video generation, AnyV2V simplifies the editing process.

---

> ### Author Response · Authors · 2024-09-19
> **Response**
>
> Thanks for your valuable comments and constructive suggestions. We just uploaded all the videos in mp4 files from the paper as Supplementary materials. We also provide comparison results among Source Video, AnyV2V, and the relevant compared methods. We have put our response to your concerns one by one in the following:
>
> > W1: Can you provide some case studies? In the four tasks mentioned in the experiment, which tasks did the AnyV2V model perform poorly? And in which scenarios did the generated videos perform poorly in these four tasks?
>
> Response to W1: Regarding the tasks that the AnyV2V model did poorly, we had a brief discussion on the Limitations (Appendix C1). Generally, our method performs poorly when the motion is too complex or fast (that the motion knowledge learned in the I2V model cannot compensate), for example in `prompt_based_failure_cases/Man Shooting a Basketball, there is a UFO in the sky/` and `prompt_based_failure_cases/purple round crystals Balls On Table/`. However, we found that in such situations other models would also perform poorly (see our failure cases in Supplementary materials). This marked the limitations of current video editing research.
>
> Among the four proposed tasks, we found that referenced-style transfer performs the worst because the I2V model lacks background guidance in style and often results in strong artifacts in the background (Refer to `style_based/a tiger walking, in the style of Vassily Kandinsky/` in Supplementary material). However, for compared methods such as TokenFlow and FLATTEN, they tend to keep minimal changes on the video, resulting in high fidelity but completely failed to apply the desired style (Refer to `style_based_failure_cases/A Border Collie Spinning Around, The Great Wave off Kanagawa style/` in Supplementary material).
>
> We will revise the appendix and add a failure case study section.
>
> > W2: In addition to the prompt-based video editing task in Tab. 2, can you compare AnyV2V, which is based on the I2V model, with other video editing models that are based on the T2I model, such as CCEdit: Creative and Controllable Video Editing via Diffusion Models, presented at CVPR 2024, in the other three tasks?
>
> Response to W2: Besides the prompt-based video editing task, We tried to compare with TokenFlow, Flatten, and Tune-A-Video in style transfer task and identity manipulation, using popular concepts (e.g. Starry Night style and popular celebrities like Elon Musk). The results are attached in Supplementary Materials.
>
> For CCEdit, we tried the officially released source code (PNP+CCEdit pipeline) and generated the result for the other three tasks except prompt-based video editing, uploaded in the Supplementary materials. Note that CCEdit is trained on conditioning the middle frame so we cannot reuse the edited images from AnyV2V.
>
> For the generated results, AnyV2V and CCEdit are both smooth. Both models show good motion following (see `subject_driven/A dog turning its head on a wooden floor/`) but sometimes with minor flicking issues in the background (See `style_based/a tiger walking, in the style of Vassily Kandinsky/` and `identity_based/Middle Aged Jack Ma Doing Exercises For The Body And Mind`). It is worth mentioning that the key difference between our work and CCEdit is that CCEdit requires pretraining on a large dataset (WebVid-10M) and thus cannot plug in pre-trained T2V models, while AnyV2V works with any off-the-shelf I2V models and we have demonstrated that with better I2V models the performance of AnyV2V would improve.
>
> > W3: Can you provide more visual results of AnyV2V in video editing? For example, in the form of videos. Also, some visual comparison results with other video editing methods would help demonstrate the performance of AnyV2V in video editing.
>
> Response to W3: We uploaded all the videos in mp4 files from the paper as Supplementary materials. Additionally, we also uploaded all baseline and comparing methods results across different tasks. We found that AnyV2V still stands out among the methods despite being a tuning-free method.
>
> (Updated 19-Sep-2024, as we corrected the CCEdit experiment for a more fair comparison.)

---

### Comment · Reviewer_oTJF · 2024-09-10
**More analysis required**

Strengths:

The introduction of AnyV2V as a novel tuning-free framework is a notable advancement in video editing research.

The framework's capability to support various video editing tasks, including prompt-based editing, reference-based style transfer, subject-driven editing, and identity manipulation, demonstrates its broad applicability.

The article presents experiments showing that AnyV2V significantly outperforms other baseline methods in both automatic and human evaluations, which is a testament to its effectiveness.  By breaking down the video editing task into a single image editing and image-to-video generation, AnyV2V simplifies the editing process.


Comments:
1. Can you provide some case studies? In the four tasks mentioned in the experiment, which tasks did the AnyV2V model perform poorly? And in which scenarios did the generated videos perform poorly in these four tasks?
2. In addition to the prompt-based video editing task in Tab. 2, can you compare AnyV2V, which is based on the I2V model, with other video editing models that are based on the T2I model, such as CCEdit: Creative and Controllable Video Editing via Diffusion Models,
presented at CVPR 2024, in the other three tasks?
3. Can you provide more visual results of AnyV2V in video editing? For example, in the form of videos. Also, some visual comparison results with other video editing methods would help demonstrate the performance of AnyV2V in video editing.

---

### Comment · Reviewer_oTJF · 2024-09-12
**More analysis required**

Strengths:

The introduction of AnyV2V as a novel tuning-free framework is a notable advancement in video editing research.

The framework's capability to support various video editing tasks, including prompt-based editing, reference-based style transfer, subject-driven editing, and identity manipulation, demonstrates its broad applicability.

The article presents experiments showing that AnyV2V significantly outperforms other baseline methods in both automatic and human evaluations, which is a testament to its effectiveness. By breaking down the video editing task into a single image editing and image-to-video generation, AnyV2V simplifies the editing process.

Comments:

Can you provide some case studies? In the four tasks mentioned in the experiment, which tasks did the AnyV2V model perform poorly? And in which scenarios did the generated videos perform poorly in these four tasks?

In addition to the prompt-based video editing task in Tab. 2, can you compare AnyV2V, which is based on the I2V model, with other video editing models that are based on the T2I model, such as CCEdit: Creative and Controllable Video Editing via Diffusion Models, presented at CVPR 2024, in the other three tasks?

Can you provide more visual results of AnyV2V in video editing? For example, in the form of videos. Also, some visual comparison results with other video editing methods would help demonstrate the performance of AnyV2V in video editing.

---

### Author Response · Authors · 2024-09-19
**Changelog**

18-Sep-2024 As suggested by Reviewer oTJF, We uploaded Supplementary Material which contains videos of AnyV2V and comparing methods.

19-Sep-2024 Modified Supplementary Material to correct CCEdit results for a fair comparison.

03-Nov-2024 Updated Camera Ready version and revised the claim regarding CLIP-score performances.

---

### Decision · Action_Editor_jjqc · 2024-10-26

**Recommendation:** Accept with minor revision

**Comment:**

The reviewers generally lean toward positive, noting that AnyV2V presents a simple yet effective tuning-free framework for video editing. The key strength highlighted is the method's broad applicability across various editing tasks without requiring additional training. While the technical implementation is straightforward, reviewers expressed concerns about limited scientific novelty, which is reflected in their weak opposition to including the work in the ICLR Journal-to-Conference track.

After the authors' responses and additional supplementary materials, it seems to me the reviewers' concerns about methodology clarity and empirical evidence were mostly addressed. The authors demonstrated the advantages of their approach over frame-by-frame editing and clarified the role of temporal feature injection. They also corrected their claims about automatic evaluation metrics.

**Audience:**

The reviews unanimously agree that the paper would be of interest to the TMLR audience in their assessments. The work addresses video editing, which is a significant area of interest in machine learning research. The paper's approach of combining existing image editing models with image-to-video models in a tuning-free framework would be particularly relevant to researchers working on video manipulation, generative models, and diffusion models.

The practical applicability of the method across various video editing tasks (prompt-based editing, reference-based style transfer, subject-driven editing, and identity manipulation) makes it valuable for both researchers and practitioners in the field. Additionally, the fact that it's open-sourced and requires no additional training makes it particularly accessible to the community.

Even though some reviewers noted limited scientific novelty, they still acknowledged that the engineering contribution and practical utility would be of interest to the audience.

**Claims And Evidence:**

The claims in the submission are generally well-supported by evidence. Initially, one reviewer marked "Claims And Evidence: No" due to a concern about the abstract's overstated claim that the method "significantly outperforms other baseline methods in automatic and human evaluations." After the authors acknowledged this discrepancy and agreed to revise the claim to specify that they "significantly outperform other baseline methods in human evaluations and achieve comparable CLIP scores," the reviewer changed their assessment to "Yes."
The authors provided comprehensive supplementary materials, including video demonstrations, comparative analyses with baseline methods, and detailed failure case studies. They also clearly demonstrated the advantages of their approach over frame-by-frame editing through visual examples. All reviewers ultimately marked "Claims And Evidence: Yes" in their final assessments.

The only minor concern was about the use of CLIP-based metrics, which were found to be less reliable than human evaluations, but this limitation was acknowledged and explained by the authors in their response.